# Peanut Shell Extract Improves Markers of Glucose Homeostasis in Diabetic Mice by Modulating Gut Dysbiosis and Suppressing Inflammatory Immune Response

**DOI:** 10.3390/nu16234158

**Published:** 2024-11-30

**Authors:** Matthew Bender, Julianna M. Santos, Jannette M. Dufour, Hemalata Deshmukh, Scott Trasti, Moamen M. Elmassry, Chwan-Li Shen

**Affiliations:** 1Department of Medical Education, Texas Tech University Health Sciences Center, Lubbock, TX 79430, USA; matthew.bender@ttuhsc.edu; 2Department of Pathology, Texas Tech University Health Sciences Center, Lubbock, TX 79430, USA; julianna.santos@ttuhsc.edu (J.M.S.); hemalata.deshmukh@ttu.edu (H.D.); 3Department of Microanatomy and Cellular Biology, Texas Tech University Health Sciences Center, El Paso, TX 79905, USA; 4Department of Cell Biology and Biochemistry, Texas Tech University Health Sciences Center, Lubbock, TX 79430, USA; jannette.dufour@ttuhsc.edu; 5Center of Excellence for Integrative Health, Texas Tech University Health Sciences Center, Lubbock, TX 79430, USA; 6Obesity Research Institute, Texas Tech University, Lubbock, TX 79401, USA; 7Laboratory Animal Resource Center, Texas Tech University Health Sciences Center, Lubbock, TX 79430, USA; scott.trasti@ttuhsc.edu; 8Department of Molecular Biology, Princeton University, Princeton, NJ 08540, USA; elmassry@princeton.edu; 9Center of Excellence for Translational Neuroscience and Therapeutics, Texas Tech University Health Sciences Center, Lubbock, TX 79430, USA

**Keywords:** peanut shell, bioactive compounds, gene microarray, microbiome, mice

## Abstract

Background/Objective: There is strong evidence that the tripartite interaction between glucose homeostasis, gut microbiota, and the host immune system plays a critical role in the pathophysiology of type 2 diabetes mellitus (T2DM). We reported previously that peanut shell extract (PSE) improves mitochondrial function in db/db mice by suppressing oxidative stress and inflammation in the liver, brain, and white adipose tissue. This study evaluated the impacts of PSE supplementation on glucose homeostasis, liver histology, intestinal microbiome composition, and the innate immune response in diabetic mice. Methods: Fourteen db/db mice were randomly assigned to a diabetic group (DM, AIN-93G diet) and a PSE group (1% wt/wt PSE in the AIN-93G diet) for 5 weeks. Six C57BL/6J mice received the AIN-93G diet for 5 weeks (control group). Parameters of glucose homeostasis included serum insulin, HOMA-IR, HOMA-B, and the analysis of pancreatic tissues for insulin and glucagon. We assessed the innate immune response in the colon and liver using a microarray. Gut microbiome composition of cecal contents was analyzed using 16S rRNA gene amplicon sequencing. Results: PSE supplementation improved glucose homeostasis (decreased serum insulin concentration, HOMA-IR, and HOMA-B) and reduced hepatic lipidosis in diabetic mice. PSE supplementation reversed DM-induced shifts in the relative abundance of amplicon sequence variants of *Enterorhabdus*, *Staphylococcus*, *Anaerotruncus*, and *Akkermansia*. Relative to the DM mice, the PSE group had suppressed gene expression levels of Cd8α, Csf2, and Irf23 and increased expression levels of Tyk2, Myd88, and Gusb in the liver. Conclusions: This study demonstrates that PSE supplementation improves T2DM-associated disorders of diabetic mice, in part due to the suppression of innate immune inflammation.

## 1. Introduction

Type 2 diabetes mellitus (T2DM) is characterized by a combination of impaired insulin secretion and insulin resistance. The incidence of T2DM is increasing at an alarming rate and expected to affect an estimated 578 million people worldwide by 2030. In addition to impaired glucose homeostasis, T2DM is often associated with hepatic dysfunction, with more than 70% of patients with T2DM having nonalcoholic fatty liver disease [1].

Excessive oxidative stress and chronic low-grade inflammation play important roles in the progression of T2DM [2,3,4,5]. Intracellular hyperglycemia increases reactive oxygen species (ROS) production, promotes advanced glycation end-product formation and the activation of protein kinase C, and enhances polyol pathway flux. ROS stimulate the generation of inflammatory mediators and adhesion molecules, oxidized low-density lipoprotein formation, and insulin resistance [2,3,4,5]. T2DM is generally accompanied by the increased production of free radicals and/or impaired antioxidant defense capabilities. In the progression of T2DM, chronic low-grade inflammation promotes the generation of inflammatory immune cells, especially macrophages [6]. Chronic low-grade metabolic inflammation causes the systematic release of cytokines, which affects peripheral tissue metabolism. Such negative effects on tissue metabolism are thought to impair glucometabolic pathways and lead to T2DM [6]. Moreover, a long-term hyperglycemic environment results in hepatocyte expansion by progressively larger intracytoplasmic lipid accumulation (lipid droplet accumulation and cell swelling) [7]. Abnormal accumulation of lipids in hepatocytes, which is caused by lipid peroxidation and the release of proinflammatory factors, can lead eventually to irreversible liver tissue damage [8]. These hepatic pathological changes, which are associated with diabetic liver injury, mainly manifest as hepatocyte steatosis, the formation of Mallory–Denk bodies, inflammatory cell infiltration, and fibrosis [9].

Gut dysbiosis is thought to be the driver of metabolic inflammation in the development of insulin resistance and T2DM [10,11,12]. Compared with lean individuals, those with T2DM have an increased proportion of *Firmicutes* and *Actinobacteria* and a decreased proportion of *Bacteroidetes*, leading to an inflammatory cascade, insulin resistance, and oxidative stress [13]. Gut dysbiosis results in increased gut permeability and the release of endotoxins, such as lipopolysaccharide (LPS, an endotoxin), into the bloodstream [14,15]. The release of LPS into circulation causes an innate inflammatory response via Toll-like receptors (TLRs) within adipose tissue [16]. When LPS binds to TLR-4, it recruits intracellular adaptor proteins, which leads to the activation of proinflammatory kinases associated with insulin resistance [17]. The immune response from the release of LPS into circulation and the translocation of gut bacteria into the pancreas are thought to contribute directly to the development of T2DM [18]. Additionally, gut microbiota changes might be involved in the development of T2DM through modulating various metabolic pathways in the host, such as immunity, energy metabolism, lipid metabolism, and amino acid metabolism [19]. Overall, there is strong evidence that the tripartite interaction between glucose homeostasis, gut microbiota, and the host immune system is a critical component in the pathophysiology of T2DM.

Dietary bioactive compounds with anti-inflammatory and antioxidant properties have great potential to improve T2DM-induced disorders, such as hyperglycemia, insulin resistance, liver steatosis, gut dysbiosis, and dysregulated innate and adaptive immunity. Among these dietary bioactive compounds, peanut shell extract (PSE, contains ~20% luteolin, a common flavonoid) has shown great potential in improving glucose homeostasis during T2DM progression. Sun et al. reported that PSE administration caused significant decreases in fasting blood glucose and improved glucose tolerance in high-fat diet/streptozotocin (STZ)-induced diabetic rats, with comparable effects to metformin [20]. In diabetic animals, luteolin attenuates T2DM-associated hyperglycemia [21], inflammation [22], and gut dysbiosis [23]. In a prospective human cohort study with 2461 T2DM individuals, the authors reported that luteolin intake is negatively associated with all-cause and cardiac mortality of T2DM individuals [24]. We reported previously that PSE improves mitochondrial function in db/db mice via suppression of oxidative stress and inflammation in the liver, brain, and white adipose tissue [25]. However, no study thus far has evaluated the effects of PSE on gut microbiota and immune inflammation in T2DM. Thus, this study aimed to investigate the effects of PSE on glucose homeostasis, gut microbiota, and immune inflammation in obese db/db diabetic mice. We hypothesized that PSE supplementation would improve glucose homeostasis, improve gut microbiota composition, and reduce innate immunity inflammation in diabetic mice. The findings of this study will elucidate how PSE might be used to better treat complications associated with T2DM.

## 2. Materials and Methods

### 2.1. Animals and Treatments

Male homozygous BKS.Cg-*Dock7^m^* +/+ *Lepr^db^*/J (db/db) mice (5-week-old, *n* = 14, Jackson Laboratory, Bar Harbor, ME, USA, strain #: 000642) were used to study T2DM. Male C57BL/6J mice (5-week-old, *n* = 6) were used for the control group (Jackson Laboratory, strain #: 000664). All animals were housed (2 mice/cage) at a constant temperature (22 ± 2 °C) and humidity (55 ± 5%) with a 12-h light/dark cycle. Mice received the AIN-93G diet (catalog number: D10012G, Research Diet, Inc., New Brunswick, NJ, USA) consisting of 59.3% carbohydrates, 18.1% protein, and 7.1% fat for 1 week. The db/db mice were then randomly assigned to either the diabetic control group (DM) or PSE group (PSE at 1.0% wt/wt in the AIN-93G diet) for 5 weeks. Both the control and DM animals received the AIN-93G diet during the 5-week study period. Food and water were provided ad libitum throughout the study period. Sabinsa Corporation (East Windsor, NJ, USA) provided our lab with PSE containing 20% luteolin concentration as a gift. The animals’ body weight, food intake, and water consumption were recorded biweekly throughout the study period. All procedures were approved by the Institutional Animal Care and Use Committee (IACUC, protocol #: 22017, approved on 27 May 2022). All experiments were conducted in accordance with the relevant guidelines and regulations.

Published work [25] indicates that to obtain significance at α = 0.05 and statistical power 0.9, *n* = 6–8 animals per group are necessary to detect differences in insulin. Thus, we used *n* = 6–7 per group for this study.

### 2.2. Sample Collection

At the end of the study, the animals were fasted for 4 h before sample collection. We collected blood from the tail vein and measured blood glucose levels using a glucometer (Accu-Check Aviva Glucose Meter, Roche Diabetes Care, Inc., Indianapolis, IN, USA). Animals were subsequently anesthetized with isoflurane and euthanized. Following euthanasia, blood samples were collected via cardiac puncture method. Blood samples were centrifuged at 1500× *g* for 20 min, and serum samples were obtained and kept at −80 °C for later analysis. Liver tissues were collected and fixed in 10% neutral buffered formalin solution, while pancreases were stored in Z-fix (AnaTech Ltd., Battle Creek, MI, USA) at room temperature for later histological assessment. The liver, colon, and cecal feces were collected, immersed in liquid nitrogen, and stored at −80 °C for later analysis.

### 2.3. Serum Insulin and Homeostasis Model Assessment

We measured serum insulin levels using an insulin ELISA kit (EMD Millipore Co., Billerica, MA, USA). The HOMA-IR (homeostasis model assessment of insulin resistance) index was calculated as [fasting serum glucose (mmol/L) × fasting serum insulin (U/L)/22.5] to assess insulin resistance. The HOMA-B (homeostasis model assessment of β-cell function) index was calculated as [(360 × fasting serum insulin (microunits/mL))/(fasting serum glucose (mmol/L) − 63)] to assess β-cell function.

### 2.4. Histology

Liver tissues were embedded in paraffin, sectioned, and stained with hematoxylin and eosin (H&E) for histological assessment. Whole sections were scanned using a ZEISS Axioscan 7 (ZEISS, St. Louis 63122, MO, USA). The slides were reviewed by a veterinary pathologist in a blinded fashion. Pancreas tissue sections were immunostained for insulin and glucagon. We used guinea pig anti-insulin (diluted 1:1000; Dako Agilent Pathology Solutions, Santa Clara, CA, USA) and mouse anti-glucagon (diluted 1:5000; Sigma-Aldrich, Inc. St. Louis 14508, MO, USA) primary antibodies (as described previously [26]) to immunostain the pancreas tissue sections for insulin and glucagon. Tissue sections were counterstained with hematoxylin.

### 2.5. Gut Microbiota Profiling Using 16S rRNA Amplicon Sequencing

Genomic DNA was extracted from the cecal contents using the PowerFecal DNA Isolation kit (Qiagen, Gaithersburg, MD, USA). Amplicon sequencing of the V4 variable region of the 16S rRNA gene was performed by MR DNA (Molecular Research LP, Shallowater, TX, USA). The V4 variable region was amplified using PCR primers 515F/806R. Samples were multiplexed and pooled together in equal proportions based on their molecular weight and DNA concentration. We then purified pooled samples using calibrated Ampure XP beads, which were then used in Illumina DNA library preparation. Sequencing was performed by MR DNA (www.mrdnalab.com, Shallowater, TX, USA) on a MiSeq following the manufacturer’s guidelines. We deposited our raw sequencing data with BioProject in the National Center for Biotechnology Information (NCBI) BioProject database.

### 2.6. Microarray Analysis

RNA from whole liver and colon tissues was purified using the Qiagen RNeasy Plus Universal Kit (cat. no. 73404) according to the manufacturer’s protocol. cDNA was synthesized using the Qiagen RT^2^ First Strand Kit (cat. No. 330401, PAMM-052ZA). DNA microarray of innate and adaptive immune response genes was performed using Qiagen RT^2^ Profiler PCR Array (cat. no. 330231 PAMM-052ZA). Fold change calculations (gene expression ratios) were calculated using the ∆∆CT method and the *p*-values were calculated based on a student’s *t*-test of the replicate 2 ∆CT values using Qiagen’s analysis software. A heatmap was constructed according to the fold regulation changes.

### 2.7. Statistical Analysis

Experimental data were given as the mean ± standard deviation (SD) or mean ± standard error of the mean (SEM). Statistical analyses of glucose homeostasis and immunity parameters were carried out by one-way analysis of variance (ANOVA) followed by *post hoc* Tukey’s test using GraphPad Prism software (version 9.0) (GraphPad Software, San Diego, CA, USA). A *p*-value lower than 0.05 was considered significant.

For the gut microbiota analysis, 16S rRNA gene sequencing data were analyzed using QIIME 2 [27]. Results of the analysis were filtered, denoised, and merged. DADA2 was used to identify exactly which amplicon sequence variants (ASVs) were present. For taxonomy assignment, the Silva database (release 138) was used [28]. To compare the relative abundance of taxa between groups, we performed compositional analysis using LOCOM, a logistic regression model for testing differential abundance in compositional microbiome data with false discovery rate control [29]. *p*-values of less than 0.05 were considered significant, unless stated otherwise. Visualization was performed in R (version 4.0.5) (codename “Shake and Throw”).

## 3. Results

### 3.1. Analysis for Insulin Resistance and Pancreatic Islet Function

The impact of PSE administration on glucose homeostasis and pancreatic islets was examined (Figure 1 and Figure 2). Compared to the control group, the DM group had higher values of serum insulin (Figure 1A), HOMA-IR (Figure 1B), and HOMA-B (Figure 1C). Supplementation of PSE significantly reduced DM-induced serum insulin and insulin resistance as shown by HOMA-IR and HOMA-B levels in db/db mice, with levels similar to the control mice (Figure 1).

Figure 2A–D (insulin) shows the histological analysis of the pancreas for insulin-producing islet beta cells, while Figure 2E–H (glucagon) shows the histological analysis for glucagon-producing alpha cells via immunohistochemistry. The control group had normal insulin and glucagon staining and normal islets. The db/db group exhibited damaged islets (combination of normal islets, islets with decreased insulin staining, and islets with damage, and β-cell loss). PSE supplementation had no effect on the pancreatic β-cells of db/db mice as shown normal islets but had lower insulin staining (Figure 2D) as well as islets with beta cell loss (Figure 2C). Consistent with this result, glucagon staining in the diabetic mice was abnormal, with the majority of islets having alpha cells localized throughout the islet (Figure 2F). Interestingly, in most of the islets, there appeared to be an increase in the number of alpha cells, although a few islets were normal or had a decrease in alpha cells (Figure 2F). Again, PSE treatment had no effect on pancreatic alpha cells compared to the diabetic mice; there were islets with an increase in the number of alpha cells (Figure 2G) and those more similar to normal islets (Figure 2H).

### 3.2. Effects of PSE Supplementation on Liver Histology

Figure 3 shows the effects of PSE supplementation on liver histology. The control liver is histologically normal (Figure 3A). The DM group had hepatic lipidosis, as shown by the diffused severe hepatocellular vacuolar change (Figure 3B). Compared to the DM group, the PSE group showed minimal to mild hepatic lipidosis (vacuolar change).

### 3.3. Effects of PSE on Intestinal Microbiome

We first examined PSE’s effect on the alpha diversity of the microbiome. Pielou’s evenness, a reflection of the abundance and evenness of the bacterial community, demonstrated no difference between the control, DM, and PSE groups (*p* > 0.05) (Figure 4). Faith’s phylogenetic diversity, reflecting the richness of the bacterial community, exhibited no difference between the control, DM, and PSE groups (*p* > 0.05) (Figure 4).

Next, we aimed to determine the differences in microbiome composition. We performed compositional microbiome analysis using LOCOM (data presented here with *p* < 0.05). While DM altered the relative abundance of several microbiome ASVs, we focused on those that were altered by PSE as well (Figure 5). We observed changes in the abundance of ASVs belonging to Actinobacteria, Firmicutes, and Verrucomicrobiota. Compared to the control group, the DM group had an increased abundance of ASVs for *Enterorhabdus*, *Staphylococcus*, and *Anaerotruncus*, while it had a decreased abundance of ASVs for *Enterorhabdus* and *Akkermansia*. Supplementation of PSE in the diet reversed all observed effects induced by DM for all five ASVs (Figure 5).

### 3.4. Results of Microarray Analysis

We reported the effect of PSE supplementation on innate immunity gene expression in liver and colon tissues (Figure 6). Utilizing a preformatted gene pathway array, we compared the expression of 84 inflammatory chemokine, cytokine, and interleukin receptor genes across the control, DM, and PSE groups. After disregarding unregulated, non-detectable gene products and presenting the relative levels of gene expression across all samples, we identified 42 inflammatory genes that categorized 23 temporally distinct groups. Then, we focused on genes that were altered in multiple comparisons across groups or tissues. The microarray analysis demonstrated significant gene regulation differences in adaptive and innate immunity between the DM and PSE groups in both colon and liver tissue (Figure 6).

Compared to the control group, the DM group had increased gene levels of Toll-like receptor 6 (Tlr6) (liver), tyrosine kinase 2 (Tyk2) (colon, liver), nuclear factor kappa B (Nfkb1) (colon, liver), beta-B-glucuronidase (Gusb) (colon, liver), and β-*actin* gene (Actb) (colon, liver), while it had decreased gene levels of CD8α (colon, liver) and colony stimulating factor 2 (Csf2) (liver). Relative to the DM mice, the PSE-supplemented db/db mice demonstrated significant decreases in the gene expression of CD8 antigen, alpha chain (Cd8α) (liver), Csf2 (liver), and interferon regulatory factor 3 (Irf3) (liver). Additionally, the PSE-supplemented mice exhibited significant increases in the gene expression of Tlr6 (colon), Tyk2 (liver), myeloid differentiation primary response gene 88 (Myd88) (colon, liver), and Gusb (liver) when compared to the DM group.

## 4. Discussion

Using the Cg-*Dock7^m^* +/+ *Lepr^db^*/J (db/db) mouse model, we examined the impacts of PSE on anti-diabetic activity in T2DM mice. Serum insulin and HOMA analysis are standard methods to measure insulin resistance and beta cell function. Elevated serum insulin is indicative of T2DM. HOMA analysis takes into account both serum insulin and fasting blood glucose and has been validated as a measure of insulin resistance and beta cell function. Insulin resistance and beta cell dysfunction are direct factors in the development of T2DM. In the present study, we found that PSE improved glucose homeostasis and insulin sensitivity, as shown by reduced serum insulin, HOMA-IR, and HOMA-β in db/db mice, which corroborates the findings of published studies [20,30,31,32]. PSE was shown to be effective in promoting insulin sensitivity, decreasing inflammation, and lowering blood glucose levels in a type 1 diabetes mouse model [20]. Additionally, luteolin (the most abundant flavonoid polyphenolic compound in PSE) has demonstrated anti-diabetic capacities that mitigate T2DM-associated disorders in animals [30,31,32]. For instance, luteolin alleviates diabetic dyslipidemia in rats by modulating Acyl-coenzyme A:cholesterol acyltransferase-2 (ACAT-2), peroxisome proliferator-activated receptor alpha (PPARα), sterol regulatory element binding protein-2 (SREBP-2) protein, and oxidative stress in rats with T2DM induced by HFD/STZ. Luteolin protects T2DM animals against diabetic cardiomyopathy by suppressing the c-JUN N-terminal kinase (JNK)-suppressed autophagy pathway [31] and improves vascular complications associated with T2DM by improving the systemic and metabolic profile in T2DM Goto-Kakizaki rats [32].

In our study, the pancreatic islet histology in the DM group was similar to previous studies in db/db diabetic mice where they reported beta cell damage and an increase in the number of alpha cells [33,34,35,36,37]. The observed increase in alpha cells could be the result of the increase in serum insulin, as insulin has been found to increase alpha cell proliferation [37]. This increase in alpha cells likely exacerbates the detrimental effects of diabetes as insulin resistance combined with increased glucagon promotes hyperglycemia and diabetes-associated metabolic dysregulation. Although we observed that PSE improved glucose homeostasis, no significant histological changes were observed in pancreatic insulin-producing islet beta cells or glucagon-producing alpha cells due to PSE treatment in db/db mice. Further study of the gene expression changes occurring within the islets should be performed in future studies to determine the molecular changes within the beta cells leading to the normalization of insulin secretion, insulin resistance, and beta cell function observed following PSE treatment.

In the present study, the histological examination of liver lipidosis in the DM group is consistent with published studies in male db/db mice [26] and HFD/STZ-induced diabetic mice [38]. This is the first study to demonstrate the mitigating effects of PSE and its bioactive compounds on hepatic lipidosis in db/db mice. PSE’s effect on liver histology works, in part, by improving liver mitochondrial biogenesis and the suppression of oxidative stress and inflammation [25]. Deshmukh et al. reported that dietary supplementation of PSE restored the T2DM-induced changes in (i) mitochondrial function-associated parameters, such as fission (DRP1), fusion (MFN1, MFN2, OPA1), biogenesis (TFAM, PGC-1α), and mitophagy (PINK1), and (ii) inflammation (TNF-α) in the liver of db/db mice [25]. Our study also aligns well with the recently growing evidence of the potential benefits of luteolin intake in treating glycolipid metabolism disorders, particularly insulin resistance, diabetes, and obesity [39].

There is much evidence which suggests that changes in the composition of gut microbiota may affect the host’s energy homeostasis, systemic inflammation, lipid metabolism, and insulin sensitivity in obesity and obesity-associated disorders, such as T2DM [40]. Changes to the gut microbiome (gut dysbiosis) can lead to dysregulation of intestinal microbial metabolites, which can trigger mechanisms that lead to insulin resistance and T2DM [41]. As a major component of the human diet, polyphenols have demonstrated the capability to modulate the composition of gut microbiota and reduce HFD-induced obesity and obesity-associated T2DM [40]. In the present study, the results show that the relative decreased abundance of *Enterohabdus* in the Eggerthellaceae family of the *Actinobacteria* phylum and *Akkermansia* in the *Akkermansiacece* family of the *Verrucomicrobia* phylum in the db/db mice is consistent with Li’s study in Zucker diabetic fatty rats [42]. The *Akkermansia* has shown to improve metabolic disorders (i.e., T2DM) via it’s anti-inflammatory effects [43]. The relative abundances of polyphenol-degrading *Enterorhabdus* and *Akkermansia* in the gut microbiota have been associated with lignin metabolism [44]. The relative increased abundance of *Akkermansia* has been reported to be associated with polyphenol-rich diets [40,45]. The finding that PSE supplementation in db/db mice significantly increased the abundance of both *Actinobacteria* and *Verrucomicrobia* is probably due to PSE’s high concentration of polyphenolic compounds. For instance, Li et al. reported that the beneficial properties of increased *Akkermansia* abundance have been shown in polyphenolic berberine-treated Zucker diabetic fatty rats [42]. Such an increase in the abundance of *Akkermansia* in the cecal feces of PSE-treated diabetic mice is supported by Gao et al. in PSE-treated depressive mice [46].

The observation that the DM mice have a significantly increased abundance of *g_Staphylococcus* and *g_Anaerotruncus* is supported by published work [47,48]. *Staphylococcus* was a common purulent coccus that predisposes hosts to a variety of purulent infections [49]. Chronic exposure to a toxin made by *Staphylococcus aureus* bacteria produces several symptoms of T2DM in rabbits, including insulin resistance, glucose tolerance, and systemic inflammation [50]. The abundance of *Streptococcaceae* was significantly increased in KK-Ay spontaneous DM mice relative to controls at the family level [47], while the critical role of *Anaerotruncus* has been demonstrated in the progression of *inflammation*, obesity, and metabolic *diseases, including T2DM* [47,48]. Liu et al. reported that the abundance of *Anaerotruncus* increased dramatically in KK-Ay spontaneous DM mice relative to controls [47]. *Anaerotruncus* promotes inflammation and undermines the integrity of the epithelial barrier due to its proinflammatory properties [51]. An HFD increases harmful bacteria (*Anaerotruncus*), reduces the abundance of dominant bacteria and beneficial bacteria such as *Lactobacillus johnsonii* and *Lactobacillus reuteri*, promotes inflammation, and damages the intestinal barrier [52]. In this study, the findings showed that PSE supplementation mitigated the T2DM-induced abundance of *g_Staphylococcus* and *Anaerotruncus* in the gut of db/db mice. Our result indicates PSE’s anti-diabetic potential via the modulation of gut dysbiosis.

T2DM negatively affects the immune system. It is the result of insulin resistance and subsequent inhibition of the insulin signaling pathway. As hyperglycemia continues, the body’s immune function is impaired and results in the increased secretion of inflammatory cytokines and the development of T2DM [53]. Evidence also shows that diabetes damages the innate and adaptive immune responses, such as impairing/inhibiting the function of natural killer cells (NK cells) and lymphocytes, thereby interfering with early defense mechanisms [54]. In the present study, we used a DNA microarray (which targets 84 of the most relevant genes associated with the innate and adaptive immune responses) to evaluate how PSE affected immune responses in both the liver and colon of db/db mice. We noted that the DM mice showed similar patterns of immune response in both the colon and liver, and that the liver exhibited greater response levels than those in the colon. Such differential responses between the colon and liver are interesting as the liver is usually perceived as a major non-immunological organ engaged primarily in metabolic, nutrient storage, and detoxification activities [55].

T2DM is considered a chronic low-grade inflammatory disease, and Nfkb controls the expression of many genes that affect inflammation and the immune response [56]. Increased Nfkb signaling may be involved in the development of insulin resistance, muscle loss, and weakness. Nfkb activation can occur in most cell types, including those in the liver, adipose tissue, and central nervous system [57]. Nfkb encodes the p105 subunit of Nfkb, which is processed to generate the NFkb1 p50 subunit. Nfkb1 is the most highly expressed transcription factor in macrophages, key cellular drivers of inflammation and immunity [58]. Nfkb1 regulates the immune response by controlling the expression of genes that release cytokines and anti-microbial molecules. Nfkb1 is a central mediator of proinflammatory gene induction and functions in both innate and adaptive immune cells. We noted that PSE supplementation suppressed DM-induced Nfkb1 gene expression in the liver of db/db mice due to PSE’s anti-inflammatory property. Such suppression of Nfkb1 gene expression in the liver of PSE-supplemented diabetic mice is consistent with Fatimawali’s study using anti-inflammatory bioactive compounds, namely *Clerodendrum minahassae* leaf extract, for managing T2DM via interacting with and inhibition of Nfkb in the insulin signal transduction pathway [59]. Moreover, Irf3, a major transcriptional regulator of adipose inflammation, is involved in maintaining systemic glucose and energy homeostasis. Elevated Irf3 gene expression has been reported in the adipocytes of obese mice and humans. Signaling through Tlr3 and Tlr4, which lie upstream of Irf3, induced insulin resistance in murine adipocytes, while Irf3 knockdown prevented insulin resistance [60]. The finding that the Irf3 gene expression level was upregulated by DM and downregulated by PSE in the liver corroborates PSE’s anti-diabetic effects via, in part, suppression of Irf3 genes.

We noted that only male mice were included in the experiment. Given that this was our initial pilot study, we chose to focus only on males, as this is more consistent with previous studies [20,25]. Future studies will incorporate female mice.

## 5. Conclusions

Dietary PSE supplementation improved glucose homeostasis and gut dysbiosis as well as mitigated liver lipidosis in diabetic mice. These improvements in T2DM-associated disorders may be due in part to suppression of the inflammatory innate immune response. The PSE used in this study is a mixture of all bioactive PSE compounds, including luteolin. Future studies are warranted to elucidate which bioactive compounds in PSE drive its anti-diabetic potential in T2DM management.

## Figures and Tables

**Figure 1 nutrients-16-04158-f001:**
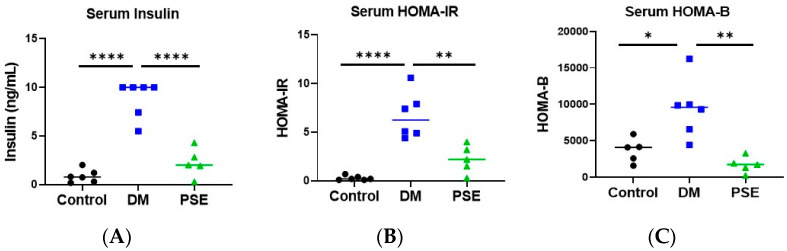
Effect of PSE supplementation on serum insulin (**A**), HOMA-IR (**B**), and HOMA-B (**C**) (*n* = 5–6 per group). Group assignment included the control group (C57BL/6J mice), DM group (db/db mice without PSE), and PSE group (db/db mice fed with 1% PSE in diet). Data was analyzed by one-way ANOVA followed by post hoc Tukey’s test. * *p* < 0.05, ** *p* < 0.005, and **** *p* < 0.00005.

**Figure 2 nutrients-16-04158-f002:**
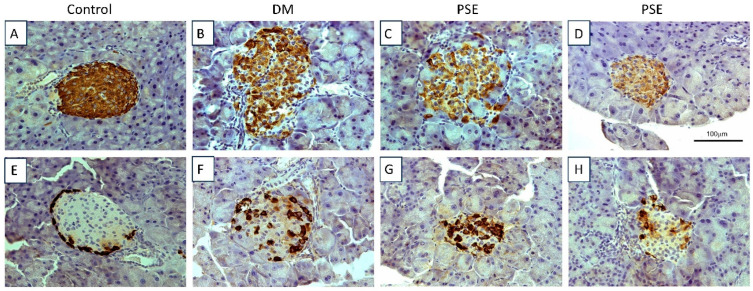
Immunohistochemical assessment of pancreatic tissue in mice after PSE supplementation. Pancreatic tissues (*n* = 6–7 per group) were sectioned and immunostained for insulin (**A**–**D**) and glucagon (**E**–**H**). All sections were counterstained with hematoxylin. Group assignment included the control group (C57BL/6J mice), DM group (db/db mice without PSE), and PSE group (db/db mice fed with 1% PSE in diet).

**Figure 3 nutrients-16-04158-f003:**
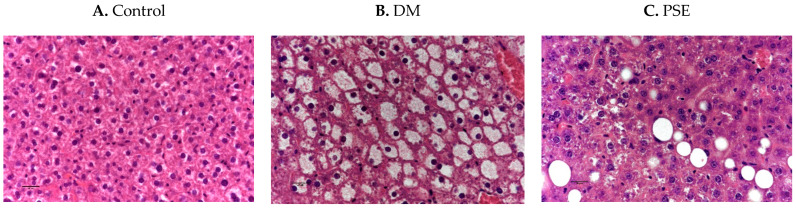
Immunobiological assessment of liver tissue in mice after PSE supplementation. Liver tissues (*n* = 6–7 per group) were sectioned and H&E stained (40×). Group assignment included the (**A**) control group (C57BL/6J mice), (**B**) DM group (db/db mice without PSE), and (**C**) PSE group (db/db mice fed with 1% PSE in diet).

**Figure 4 nutrients-16-04158-f004:**
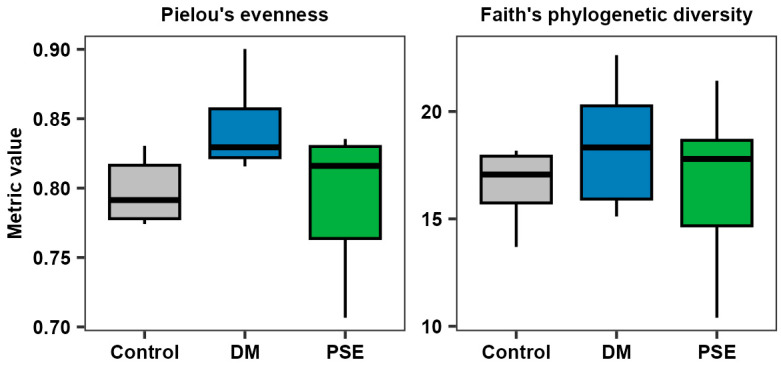
Alpha diversity of the gut microbiome. Group assignment included the control group (C57BL/6J mice), DM group (db/db mice without PSE), and PSE group (db/db mice fed with 1% PSE in diet). *n* = 6–7 per group. The Kruskal–Wallis test was performed, followed by Dunn’s test, to determine statistical significance. No difference was observed across groups.

**Figure 5 nutrients-16-04158-f005:**
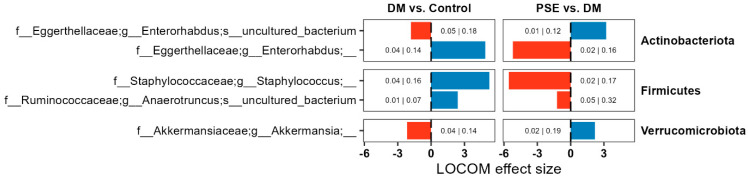
Changes in the microbiome composition across groups. Group assignment included the control group (C57BL/6J mice), DM group (db/db mice without PSE), and PSE group (db/db mice fed with 1% PSE in diet). Reported ASVs are restricted to those altered by both DM and PSE. LOCOM was used to determine statistical significance. Data presented here with *p* < 0.05, but adjusted *p* values are denoted as well (*p*-value|Adjusted *p*-value).

**Figure 6 nutrients-16-04158-f006:**
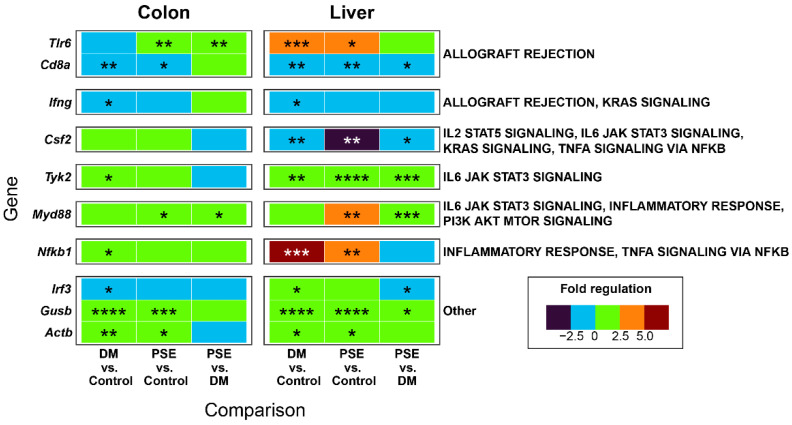
Effects of PSE supplementation on innate immunity microarray. *n* = 6–7 per group. * *p* < 0.05, ** *p* < 0.01, *** *p* < 0.001, and **** *p* < 0.0001.

## Data Availability

The original contributions presented in the study are included in the article, further inquiries can be directed to the corresponding author.

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
