# Peer review of "Peanut Shell Extract Improves Markers of Glucose Homeostasis in Diabetic Mice by Modulating Gut Dysbiosis and Suppressing Inflammatory Immune Response"

_nutrients, 2024, doi:10.3390/nu16234158_

Round 1
Reviewer 1 Report
Comments and Suggestions for Authors
Bender and colleagues demonstrated that PSE supplementation positively impacts glucose homeostasis, reduces hepatic lipidosis, and alters gut microbiome composition in diabetic mice. The treatment also modulated immune response genes, lowering liver inflammation. These findings indicate that PSE could potentially mitigate T2DM-related complications by targeting metabolic, microbiome, and immune pathways.
The manuscript is interesting and well-suited for publication; however, the authors should address a few minor points for clarity:
- Why were only male mice included in the experiment? Could the authors comment on the exclusion of females?
- How was PSE administered to the mice—via diet or injection? Please elaborate on the composition of the DM + PSE group.
- Please add error bars to Figures 2 and 3.
- In Figure 3, a magnified image of the H&E-stained sections would make the lipidosis details clearer.
Author Response
Reviewer #1:
Bender and colleagues demonstrated that PSE supplementation positively impacts glucose homeostasis, reduces hepatic lipidosis, and alters gut microbiome composition in diabetic mice. The treatment also modulated immune response genes, lowering liver inflammation. These findings indicate that PSE could potentially mitigate T2DM-related complications by targeting metabolic, microbiome, and immune pathways.
The manuscript is interesting and well-suited for publication; however, the authors should address a few minor points for clarity:
- Why were only male mice included in the experiment? Could the authors comment on the exclusion of females?
Response: Thanks for the comment. Given that this was our initial pilot study, we chose to focus on only males, as this is more consistent with published previous studies (Deshmukh 2024, Sun 2018). Future studies will incorporate female mice. We have added the following at the end of Discussion (line 392-394):
“We noted that only male mice included in the experiment. Given that this was our initial pilot study, we chose to focus on only males, as this is more consistent with published previous studies (20, 25). Future studies will incorporate female mice.”
- How was PSE administered to the mice—via diet or injection? Please elaborate on the composition of the DM + PSE group.
Response: 1% PSE was incorporated into the AIN-93 diet and provided to animals (line 108).
- Please add error bars to Figures 2 and 3.
Response: Thanks for the comments. In this R1, the error bars/scale have been added to Figures 2 and 3, accordingly.
- In Figure 3, a magnified image of the H&E-stained sections would make the lipidosis details clearer.
Response: Thanks for the comments. In this R1, we have revised Figure 3 accordingly.
Reviewer 2 Report
Comments and Suggestions for Authors
The findings of this study underscore the beneficial effects of peanut shell extract (PSE) supplementation on type 2 diabetes mellitus (T2DM) in db/db mice. PSE not only improved glucose homeostasis by decreasing serum insulin levels and enhancing insulin sensitivity, but it also mitigated hepatic lipidosis, suggesting a protective role against liver damage associated with diabetes. Additionally, PSE supplementation positively influenced the gut microbiome, reversing harmful shifts in specific bacterial populations. The alterations in immune response observed in the liver, characterized by changes in gene expression related to inflammation, further support the notion that PSE may exert its effects through the modulation of innate immune pathways. Overall, this study highlights the potential of PSE as a therapeutic agent for managing T2DM-related complications, emphasizing the importance of the interplay between diet, gut microbiota, and immune function in glucose metabolism.
comments:
1. affliction number sequence please write in sequence,
2. discuss more about the selection of insulin and HOMA
3. Figure 6 the font size and reshaped.
4. how to prove the cause effect relationship of T2DM and PSE, as some parameters maybe caused by T2DM itself rather than the dietary PSE?
Author Response
Reviewer #2:
The findings of this study underscore the beneficial effects of peanut shell extract (PSE) supplementation on type 2 diabetes mellitus (T2DM) in db/db mice. PSE not only improved glucose homeostasis by decreasing serum insulin levels and enhancing insulin sensitivity, but it also mitigated hepatic lipidosis, suggesting a protective role against liver damage associated with diabetes. Additionally, PSE supplementation positively influenced the gut microbiome, reversing harmful shifts in specific bacterial populations. The alterations in immune response observed in the liver, characterized by changes in gene expression related to inflammation, further support the notion that PSE may exert its effects through the modulation of innate immune pathways. Overall, this study highlights the potential of PSE as a therapeutic agent for managing T2DM-related complications, emphasizing the importance of the interplay between diet, gut microbiota, and immune function in glucose metabolism.
comments:
- affliction number sequence please write in sequence.
Response: Thanks for the comment. In this R1, we have corrected the affiliation number sequence.
- discuss more about the selection of insulin and HOMA
Response: Thanks for the comment. In this R1, we have included the following in Discussion:
“Serum insulin and HOMA analysis are standard methods to measure insulin resistance and beta cell function. Elevated serum insulin is indicative of T2DM. HOMA analysis takes into account both serum insulin and fasting blood glucose and has been validated as a measure of insulin resistance and beta cell function. Insulin resistance and beta cell dysfunction are direct factors in development of T2DM. In the present study”.
- Figure 6 the font size and reshaped.
Response: Thanks for the comments. In this R1, we have revised Figure 6 accordingly.
- how to prove the cause effect relationship of T2DM and PSE, as some parameters maybe caused by T2DM itself rather than the dietary PSE?
Response: This study included control, DM and PSE treated groups. These groups allow the comparison of animals not treated with PSE without diabetes and with diabetes, which can be used to determine the effects associated with diabetes. The PSE group was treated the same as the DM group and therefore, the changes observed with the PSE group are associated with PSE treatment.